# Effects of 2,2′-Azobis(2-amidinopropane) dihydrochloride (AAPH) on Functional Properties and Structure of Winged Bean Protein

**DOI:** 10.3390/foods14234120

**Published:** 2025-12-01

**Authors:** Wei Fang, Jianglin Li, Zhaoxia Qu, Jiabin Hu, Dongming Chen, Xingjian Huang

**Affiliations:** 1College of Biological and Food Engineering, Huaihua University, Huaihua 418000, China; fangwei@hhtc.edu.cn (W.F.); linzia2010@163.com (J.L.); quzhaoxia@hhtc.edu.cn (Z.Q.); hujiabin202303@163.com (J.H.);; 2Hunan Provincial Higher Education Key Laboratory of Intensive Processing Research on Mountain Ecologcal Food, Huaihua 418000, China; 3Key Laboratory of Research and Utilization of Ethnomedicinal Plant Resources of Hunan Province, Huaihua 418000, China

**Keywords:** AAPH, winged bean protein, oxidative modification, functional properties, structure

## Abstract

**Background:** The impact and regulation of protein oxidative modification on protein functional properties is a research hotspot in food processing. This study aimed to clarify the mechanism of free radical oxidation on the structure and function of winged bean protein. **Methods:** Winged bean protein was treated with different concentrations of AAPH (0.04 mmol/L, 0.20 mmol/L, 1.00 mmol/L). The functional properties (solubility, surface hydrophobicity, zeta potential), oxidation degree indicators, and secondary and tertiary structures of winged bean protein were tested and characterized under different oxidation conditions. **Results:** Low-concentration (0.04 mmol/L) AAPH led to the decomposition of winged bean protein, with decreased particle size and increased surface hydrophobicity and solubility; medium-concentration (0.20 mmol/L) AAPH caused significant aggregation of winged bean protein, with decreased surface hydrophobicity and solubility; high-concentration (1.00 mmol/L) AAPH led to the rearrangement of winged bean protein aggregates, forming more soluble aggregates and increasing solubility. With the gradual increase in AAPH addition, the α-helix and random coil structures of winged bean protein showed a trend of first increasing and then decreasing, while the β-sheet structure showed a trend of first decreasing and then increasing, and the β-turn structure remained almost unchanged. **Conclusions:** Under mild oxidation conditions (AAPH = 0.04 mmol/L), the functional properties of winged bean protein could be optimized. However, under relatively strong oxidation conditions (AAPH > 0.20 mmol/L), the structural integrity and functionality of winged bean protein would be compromised. This study helps deepen our understanding of the oxidative modification mechanism of winged bean protein.

## 1. Introduction

Winged bean (*Psophocarpus tetragonolobus* (L.) DC.), also known as carambola bean, dragon bean, or emperor bean, is currently the root crop with the highest protein content in the world. Winged bean, with its remarkable nutritional value and development potential, stands out due to its high protein content, with seed protein accounting for approximately 30% to 40% of total dry matter. With the growing global demand for plant-based proteins, winged bean protein, as a sustainable source of plant protein, has attracted increasing attention for its functional properties and application potential. According to previous studies, winged bean protein possessed many excellent physicochemical properties similar to soybean protein, and even surpassed soybean protein in many aspects [1,2]. Current research on winged bean protein mainly focuses on basic physicochemical properties and nutritional composition analysis. There is still a lack of an in-depth understanding of the relationship between the functional properties and the structure of winged bean protein. Therefore, it is necessary to conduct in-depth research on the relationship between the structure and function of winged bean protein.

Protein oxidation was a common issue in food processing and storage, which affected the structure, functional properties, and nutritional value of proteins [3,4,5,6]. For instance, oxidation could lead to protein aggregation and burial of cleavage sites, thereby reducing protein digestibility in vitro. Furthermore, oxidative modification could lead to changes in protein secondary and tertiary structures, formation of disulfide bonds and intramolecular hydrogen bonds, as well as hydrophobic interactions, which, in turn, affected protein solubility, emulsification stability, and other functional properties [7]. Excessive protein oxidation could also reduce most functional properties, such as the activity of glutamine synthetase and Cu, Zn-superoxide dismutase [8,9].

2,2′-Azobis(2-amidinopropane) dihydrochloride (AAPH) was a commonly used hydrophilic free radical initiator. When heated in aqueous solution, it generates alkoxyl radicals and peroxyl radicals at a constant and measurable rate, often used to simulate in vitro oxidative stress environments [10,11]. AAPH, when heated, decomposed to generate alkoxy/peroxy radicals, which attacked the side chain amino acids of proteins and initiated oxidation reactions. AAPH-induced oxidative stress has been proven to have significant effects on various proteins, such as whey protein [7], peanut protein [5], egg white protein [12], shrimp myofibrillar protein [13], and walnut protein [10,14], leading to increased carbonyl content, decreased sulfhydryl content, and changes in their secondary structure and thermal stability.

Currently, there are few reports on the effects of AAPH on winged bean protein. Considering the great potential of winged bean protein as an emerging plant protein source, it was important to conduct in-depth research on the effects of AAPH oxidation on the structural and functional properties of winged bean protein. This study aimed to investigate the effects of AAPH-induced oxidation at different concentrations on the functional properties (protein solubility, surface hydrophobicity, zeta potential, and particle size distribution) and secondary and tertiary structures of winged bean protein, seeking to establish the correlation between protein structure and function. This research would provide a theoretical basis for understanding the changes in winged bean protein under oxidative conditions and provide data support for its application in the food industry, thereby promoting the development and utilization of winged bean protein in plant-based food products.

## 2. Materials and Methods

### 2.1. Materials

Winged bean seeds were sourced from Huaihua Yafan Agricultural Products Co., Ltd., located in Hunan Province, Huaihua, China. 2,2′-Azobis(2-methylpropionamidine) dihydrochloride (AAPH) was procured from Wako Pure Chemical Industries, Ltd. based in Osaka, Japan. Both 1-anilino-8-naphthalene sulfonate (ANS) and 5,5′-dithiobis (2-nitrobenzoic acid) (DTNB) were acquired from Sigma Chemical Company (St. Louis, MO, USA). All other chemical reagents were supplied by either Sigma-Aldrich Co. (St. Louis, MO, USA) or Fisher Scientific Co. (Pittsburgh, PA, USA). Every reagent utilized in this study met the standards of analytical grade or biochemical grade purity.

### 2.2. Preparation of Lipid-Free Winged Bean Protein Isolate

Lipid-free winged bean protein isolate (WPI) was prepared following a modified protocol derived from previously reported methods [15,16]. To minimize oxidative modification of the initial WPI and obtain a low-oxidation WPI sample, defatted winged bean seed flour was prepared via a laboratory-specific procedure. First, winged bean seeds were cleaned, frozen at −18 °C for 12 h, and subsequently freeze-dried for 72 h. The dried seeds were dehulled and soaked in deoxygenated water for 12 h with constant stirring; after vacuum filtration, the wet seed material was homogenized three times with five volumes of chilled acetone at −20 °C. The resulting solid residue was vacuum-dried at 20 °C and defatted three times with n-hexane (solid-to-liquid ratio 1:4, *w*/*v*) at 20 °C.

The defatted seed residue was ground to pass through an 80-mesh sieve (0.198 mm aperture) and then incubated with a mixture of n-hexane and ethanol (1:2:4, *w*/*v*/*v*) at 4 °C for 1 h. Following vacuum filtration, the filter cake was immersed in 95% (*v*/*v*) ethanol (flour-to-solvent ratio 1:5, *w*/*v*) at 20 °C for 1 h, then vacuum-filtered again and dried under vacuum at 20 °C. The dried material was re-ground to 80-mesh fineness and stored at 4 °C until further use.

WPI extraction from the defatted flour proceeded as follows: the aqueous alcohol-washed winged bean seed flake precipitate was suspended in distilled water at a ratio of 1:15 (*w*/*v*, flake to water), and the pH was adjusted to 9.0 using 2 mol/L NaOH. After stirring at 20 °C for 1 h, the suspension was centrifuged at 15,900× *g* for 30 min at 4 °C, and the supernatant was collected. WPI was precipitated by adjusting the supernatant pH to 4.2 with 2 mol/L HCl, followed by centrifugation at 6000× *g* for 30 min at 4 °C. The protein curd was rinsed with distilled water, then re-suspended in distilled water (1:5, *w*/*v*, precipitate to water) and neutralized to pH 7.0 with 2 mol/L NaOH. To remove trace insoluble impurities, the suspension was centrifuged at 15,000× *g* for 30 min at 4 °C; the resulting supernatant was freeze-dried and stored at 4 °C for subsequent experiments.

### 2.3. Protein Oxidation Treatment

A solution of winged bean protein isolate (10 mg/mL) was prepared by dissolving the protein in 0.01 mol/L sodium phosphate buffer (pH 7.4) supplemented with 0.5 mg/mL sodium azide. This protein solution was mixed with gradient concentrations of AAPH to achieve final AAPH concentrations of 0.00 (control), 0.04, 0.20, and 1.00 mmol/L, respectively. The mixtures were incubated at 37 °C for 24 h in the dark with constant orbital shaking under atmospheric conditions to induce protein oxidation. The oxidation reaction was terminated promptly by immersing the reaction vessels in an ice bath to rapidly cool the solutions to 0–4 °C. To remove trace insoluble aggregates formed during the cooling process, the protein solutions were centrifuged at 10,000 × g for 1 h at 4 °C. The resulting supernatants were dialyzed against deionized water at 4 °C for 72 h (with water refreshed every 12 h) to eliminate residual AAPH and its oxidative byproducts. Following dialysis, the oxidized winged bean protein solutions were freeze-dried and stored at 4 °C until further analysis.

### 2.4. Determination of Protein Solubility

Add 100 mL of ultrapure water (pH 7.2) to a beaker, followed by slow addition of 0.2 g protein sample. Stirred the mixture for 1 h in a thermostatic magnetic stirrer (at 40 °C and 500 r/min). Then, centrifuged the mixture in a refrigerated centrifuge (at 10,000× *g* and 25 °C chamber temperature) for 20 min. Collected the supernatant and determined the protein content using the micro-Kjeldahl method. The solubility was expressed as the ratio of protein content in the supernatant to the total protein content [17].

### 2.5. Surface Hydrophobicity (H_0_) Measurement

Freeze-dried protein samples were reconstituted in 0.01 mol/L phosphate buffer (pH 7.0) to prepare a set of serial dilutions with final protein concentrations of 0.05, 0.10, 0.20, 0.50, 1.00, and 2.00 mg/mL, respectively. For each dilution, 5 mL of the protein solution was mixed thoroughly with 40 μL of 1-anilino-8-naphthalene sulfonate (ANS) working solution (8.0 mmol/L, dissolved in 0.01 mol/L phosphate buffer at pH 7.0). Fluorescence intensity (FI) was quantified using a Hitachi F-4600 fluorescence spectrophotometer (Hitachi Limited, Tokyo, Japan) with excitation and emission wavelengths set at 390 nm and 470 nm, respectively. The H_0_ value was defined as the initial slope of the linear regression curve generated by plotting fluorescence intensity against protein concentration [18].

### 2.6. Zeta Potential Analysis

Freshly prepared winged bean protein samples were diluted to a final concentration of 0.1 mg/mL using 0.1 mol/L HCl solution adjusted to pH 2.0. The zeta potential of each sample was analyzed in three replicate measurements using a Zetasizer Nano ZS instrument (Malvern Panalytical Ltd., Worcestershire, UK), with the detection mode specifically set to zeta potential measurement.

### 2.7. Dynamic Light Scattering (DLS) Measurement

Freshly prepared winged bean protein samples were diluted to a final concentration of 0.1 mg/mL using hydrochloric acid (HCl) solution adjusted to pH 2.0. Each diluted test solution was transferred to a disposable polyacrylate cuvette with a 1 cm × 1 cm light path (model: DTS0012, Malvern Panalytical Ltd.). Dynamic light scattering (DLS) measurements were carried out at a constant temperature of 25 °C using a Zetasizer Nano ZS instrument (Malvern Panalytical Ltd., Worcestershire, UK), with the detection angle set to 173° backscattering mode. The particle size distribution of the protein samples was characterized by the volume-weighted mean particle diameter, and all measurements were performed in three independent replicates to ensure data reliability.

### 2.8. Quantification of Protein Carbonyl Content

The level of protein carbonylation in winged bean protein samples was quantified via a modified protocol based on the method reported by Fuentes-Lemus and co-workers [19]. This approach relies on the specific derivatization reaction between 2,4-dinitrophenylhydrazine (DNPH) and carbonyl moieties present in both native and oxidized protein samples. Carbonyl content was calculated as micromoles of carbonyl groups per gram of soluble protein, using a molar extinction coefficient of 22,000 M^−1^ cm^−1^ for the DNPH-carbonyl adduct. The concentration of soluble protein in each sample was determined via the bicinchoninic acid (BCA) assay, with bovine serum albumin (BSA) serving as the calibration standard.

### 2.9. Determination of Sulphydryl and Disulphide Group Contents

The concentrations of sulphydryl groups (including free and buried -SH groups) and total disulphide/sulphydryl moieties in winged bean protein samples were determined following a modified version of Ellman’s assay, as adapted by Fuentes-Lemus et al. [19]. The concentration of soluble protein in each sample was quantified via the bicinchoninic acid (BCA) method, with bovine serum albumin (BSA) used as the reference standard for calibration. The content of sulphydryl groups was calculated as micromoles of -SH groups per gram of soluble protein, utilizing a molar extinction coefficient of 13,600 M^−1^ cm^−1^ for the reaction product of Ellman’s reagent with -SH groups.

### 2.10. Analysis of Circular Dichroism (CD) Spectra

Far-ultraviolet (Far-UV) circular dichroism spectra (190–250 nm) of winged bean protein samples were acquired at 25 °C using a J-1500 circular dichroism spectropolarimeter (Jasco Corporation, Tokyo, Japan). Protein sample supernatants (prepared as described in Section 2.6) were diluted to a final concentration of 0.15 mg/mL and analyzed in 0.1 cm pathlength quartz CD cuvettes. The instrumental parameters were set as follows: a scan rate of 50 nm/min, a response time of 4 s, and a bandwidth of 1.0 nm [20]. Each recorded spectrum represented the average of three consecutive scans and was baseline-corrected by subtracting the spectrum of the protein-free buffer blank. The relative proportions of the four secondary structural components (α-helix, β-sheet, β-turn, and random coil) were calculated using Yang’s equation, with an average amino acid residue value of 110 incorporated into the computational model.

### 2.11. Analysis of Intrinsic Fluorescence Spectra

Intrinsic fluorescence spectra of winged bean protein samples were measured at 25 °C using a Hitachi F-4600 fluorescence spectrophotometer (Hitachi Limited, Tokyo, Japan), following a protocol adapted from the method reported by Ni and colleagues [21]. Protein sample supernatants were diluted to a final concentration of 0.075 mg/mL using deionized distilled water (DDW), and the diluted solutions were excited at a wavelength of 290 nm. Emission spectra were collected over a wavelength range of 300 to 400 nm. The instrumental parameters were configured as follows: excitation (Ex) and emission (Em) slit widths were both set to 5 nm, and the spectral scan rate was adjusted to 240 nm per minute.

### 2.12. Analysis of FT-Raman Spectra

Fourier-transform Raman (FT-Raman) spectra of lyophilized winged bean protein powder samples were acquired using an INVIA laser Raman spectrometer (Renishaw plc, Gloucestershire, UK) with the following instrumental parameters: excitation laser wavelength of 1064 nm, laser power output of 1 W, spectral resolution of 4 cm^−1^, and a total of 800 cumulative scans. All acquired spectra were normalized against the phenylalanine characteristic band at approximately 1004 cm^−1^ (corresponding to symmetric aromatic ring breathing vibration), which was selected as the internal standard due to its insensitivity to both protein conformational alterations and microenvironmental fluctuations. Additionally, quantitative analysis of the disulfide bond conformations in the protein samples under the experimental conditions was conducted via curve-fitting using Peakfit 4.12 software (Seasolve Software Inc., Framingham, MA, USA) [22].

### 2.13. Statistical Analysis

All experimental assays were carried out in three independent replicate measurements, and the experimental results were reported as mean values ± standard deviation (SD). Data visualization (including all figures presented in this study) was created using Origin 2018 software (OriginLab Corporation, Northampton, MA, USA). To determine the statistical significance of variations in mean values across different experimental groups, one-way analysis of variance (ANOVA) was performed, followed by Duncan’s multiple comparison test (significance level set at *p* < 0.05). All statistical analyses were conducted using SPSS Statistics version 25.0 (IBM Corporation, Armonk, NY, USA).

## 3. Results and Discussion

### 3.1. Impacts of Peroxyl Radical-Induced Oxidative Modification on the Functional Attributes of Winged Bean Protein

#### 3.1.1. Solubility

Winged bean protein was a plant protein whose solubility was affected by various factors, including pH, ionic strength, temperature, and redox environment [23].

As shown in Figure 1, when the AAPH concentration was 0.00 mmol/L, 0.04 mmol/L, 0.2 mmol/L, and 1.00 mmol/L, the solubility of winged bean protein was 86.2 ± 0.4%, 88.5 ± 0.3%, 83.5 ± 0.4%, and 88.5 ± 0.4%, respectively. When the AAPH concentration was 0.00 mmol/L, 0.04 mmol/L, and 1.00 mmol/L, the solubility of winged bean protein showed little difference, ranging from 86.2% to 88.5%, indicating that AAPH had no significant effects on the solubility of winged bean protein. Notably, when the AAPH concentration was 0.2 mmol/L, the solubility of winged bean protein decreased slightly to 83.5% (*p* < 0.05). This decrease in solubility might be related to protein aggregation, conformational changes, surface hydrophobicity, and Zeta potential. According to the subsequent further test results of this experiment, when the concentration of AAPH reached 0.2 mmol/L, the solubility of wing bean protein decreased, which was mainly due to the formation of more insoluble aggregates.

#### 3.1.2. Surface Hydrophobicity

Surface hydrophobicity reflected the number of nonpolar groups exposed on the protein molecule surface, and the exposure or concealment of these groups directly affected the protein’s interaction with water molecules or other proteins [24]. Protein oxidative modification, such as peroxyl radicals induced by AAPH, could cause changes in protein structure, thereby affecting its solubility and surface hydrophobicity [25,26]. As shown in Figure 2, when the AAPH addition levels were 0.00 mmol/L, 0.04 mmol/L, 0.2 mmol/L, and 1.00 mmol/L, the surface hydrophobicity of winged bean protein were 847.48 ± 34.22, 2451.70 ± 43.92, 1367.60 ± 55.37, and 503.69 ± 68.11, respectively. Overall, with the increase in AAPH addition, the surface hydrophobicity of winged bean protein showed a trend of first increasing and then decreasing (*p* < 0.05).

When the concentration of AAPH was 0.04 mmol/L, the surface hydrophobicity of winged bean protein significantly increased. This was because under the action of lower concentrations of AAPH, the winged bean protein underwent moderate oxidation, leading to moderate unfolding of the protein structure and exposure of internal hydrophobic groups. This moderate unfolding increased the contact area between the protein and solvent, thereby enhancing surface hydrophobicity [27]. Although increased surface hydrophobicity was usually associated with decreased solubility, proteins could also form soluble aggregates or undergo conformational changes that facilitate interactions with water molecules, resulting in increased solubility [24]. When the concentration of AAPH reached 0.20 mmol/L, the degree of oxidation intensified with increasing AAPH concentration. Oxidative stress could lead to a reduction in protein sulfhydryl (-SH) groups, promote the formation or breakage of disulfide bonds, and subsequently trigger protein aggregation and denaturation [25,28]. The aggregation of winged bean protein might re-bury previously exposed hydrophobic regions, leading to a decrease in surface hydrophobicity [15].

When the AAPH concentration was 1.00 mmol/L, under higher concentration oxidation conditions, some hydrophobic amino acid residues (such as methionine and tyrosine) might be oxidized in addition to causing aggregation, thereby reducing their hydrophobicity and even converting into hydrophilic groups [15]. For example, although the oxidation of tyrosine residues to form dityrosine was a cross-linking process, its polarity might change, leading to a decrease in surface hydrophobicity measurement results [25,27]. Furthermore, the formed aggregates were not completely dense and insoluble, but rather aggregates with looser structures or exposed internal hydrophilic groups, resulting in increased solubility.

#### 3.1.3. Zeta Potential

Zeta potential was a crucial indicator for measuring the stability of colloidal systems. The greater its absolute value, the stronger the electrostatic repulsion between particles and the more stable the system [29]. As shown in Figure 3, when the AAPH addition levels were 0.00 mmol/L, 0.04 mmol/L, 0.2 mmol/L, and 1.00 mmol/L, the Zeta potential values of winged bean protein were −18.5 ± 0.5 mV, −15.3 ± 0.2 mV, −18.8 ± 0.3 mV, and −20.5 ± 0.7 mV, respectively. Overall, with the increase in AAPH addition, the absolute value of winged bean protein’s Zeta potential showed a trend of first decreasing and then increasing (*p* < 0.05).

When the AAPH concentration increased from 0.00 mmol/L to 0.04 mmol/L, the absolute value of the zeta potential of winged bean protein decreased from 18.53 mV to 15.3 mV, indicating a reduction or neutralization of surface negative charges. This might be related to the modification or altered exposure of charged amino acid residues (such as lysine, histidine, and arginine) during oxidative modification [6]. Subsequently, when the AAPH concentration continued to increase to 0.2 mmol/L and 1.00 mmol/L, the absolute value of the zeta potential of winged bean protein gradually increased, reaching 18.77 mV and 20.47 mV, respectively. The re-increase in the absolute value of zeta potential suggested that at higher oxidation levels, the protein structure further unfolds, exposing more negatively charged groups (such as glutamic acid and aspartic acid), or due to protein aggregation forming larger particles, which changed the surface charge distribution, leading to an increase in overall negative charges [30].

### 3.2. Impacts of Peroxyl Radical-Induced Oxidative Modification on Aggregation of Winged Bean Protein

#### 3.2.1. Mean Particle Size

The change in particle size directly reflected the protein’s aggregation or deaggregation behavior [31,32]. As shown in Figure 4, when the AAPH concentration was 0.00 mmol/L, 0.04 mmol/L, 0.2 mmol/L, and 1.00 mmol/L, the average particle size of winged bean protein was 199.2 ± 4.9 nm, 129.8 ± 6.8 nm, 244.9 ± 5.4 nm, and 256.8 ± 12.6 nm, respectively. Overall, the average particle size of winged bean protein showed a gradual increasing trend with increasing AAPH concentration (*p* < 0.05).

AAPH-induced oxidation was a continuous process, and its effects on protein structure and function were not monotonically linear. At lower AAPH concentrations (0.04 mmol/L), winged bean protein underwent moderate depolymerization and conformational rearrangement, resulting in a decrease in average particle size (from 199.23 nm to 129.77 nm) and an increase in solubility (from 86.2% to 88.5%). As AAPH concentration increased to 0.2 mmol/L and 1.00 mmol/L, proteins might experience more profound oxidative damage. The average particle size increased significantly (from 129.77 nm to 244.93 nm and 256.77 nm, respectively), indicating protein aggregation. At 1.00 mmol/L, solubility increased again (to 88.5%), which might be due to partial protein fragmentation or formation of smaller, more hydrophilic oxidation products that underwent extreme oxidation conditions, or some rearrangement of the aggregate internal structure, leading to higher apparent solubility at the macroscopic level.

The physicochemical properties of winged bean protein exhibited complex nonlinear relationships during oxidation. Changes in surface hydrophobicity, Zeta potential, and average particle size collectively drove the dynamic changes in solubility [33,34]. This dynamic synergistic variation in multiple parameters was a typical characteristic of protein oxidative modification.

#### 3.2.2. Particle Size Distribution

The impact of 2,2-azobis(2-amidinopropane) dihydrochloride (AAPH)-induced oxidation on protein properties, particularly solubility, surface hydrophobicity, Zeta potential, and particle size distribution, was a significant research area that helped deepen our understanding of protein behavior changes during processing and storage [10,14].

As shown in Figure 5, when AAPH = 0.00 mmol/L, the winged bean protein showed a single particle size distribution peak with a peak position at 220 nm (light intensity ratio of 12%), and a peak width of 512.8 nm. This indicated that the protein mainly existed as monomers or small-sized aggregates. When AAPH = 0.04 mmol/L, the winged bean protein exhibited two particle size distribution peaks: the peak of the smaller particle size distribution was at 164 nm (light intensity ratio of 8.79%), with a peak width of 701.9 nm; the peak of the larger particle size distribution was at 5560 nm (light intensity ratio of 0.775%), with a peak width of 2470 nm. This suggested that the addition of low-concentration AAPH led to both significant degradation of winged bean protein, forming smaller degradation products, and self-assembly, leading to the formation of larger aggregates.

When AAPH = 0.20 mmol/L, winged bean protein exhibited two particle size distribution peaks. The peak of the smaller particle size distribution corresponded to a diameter of 295 nm (light intensity ratio of 11%), with a peak width of 800.6 nm; the peak of the larger particle size distribution corresponded to a diameter of 5560 nm (light intensity ratio of 1.51%), with a peak width of 760 nm. Compared to 0.04 mmol/L AAPH, the smaller particle size peak increased, and the light intensity ratio of the larger particle size peak increased, indicating a further increase in the proportion of large aggregates. When AAPH = 1.00 mmol/L, winged bean protein also showed two particle size distribution peaks. The peak of the smaller particle size distribution corresponded to a diameter of 295 nm (light intensity ratio of 11.5%), with a peak width of 668.2 nm; the peak of the larger particle size distribution corresponded to a diameter of 5560 nm (light intensity ratio of 2.15%), with a peak width of 760 nm. The light intensity ratio of the large aggregate peak further increased, indicating that continuous oxidation promotes the formation of larger-scale aggregates.

### 3.3. Characterization of the Oxidative Markers of AAPH-Mediated Winged Bean Protein Oxidation

The carbonyl content was an important indicator of protein oxidation degree [35,36]. Experimental results showed that the carbonyl content of winged bean protein increased with the addition of AAPH. As shown in Table 1, when the addition amounts of AAPH were 0.00 mmol/L, 0.04 mmol/L, 0.2 mmol/L, and 1.00 mmol/L, the carbonyl contents of winged bean protein were 1.75 ± 0.11 μmol/g, 2.17 ± 0.12 μmol/g, 3.06 ± 0.09 μmol/g, and 4.22 ± 0.13 μmol/g, respectively. This indicated that higher AAPH concentration led to more severe oxidative modification and more carbonylated products inside the protein. This finding was consistent with multiple studies. The oxidation degree of whey protein isolate induced by AAPH increased significantly with the increase in AAPH concentration, as shown by higher protein carbonyl levels [37]. Moreover, the carbonyl content of arachin and egg white protein also increased significantly with increasing AAPH concentration [12,38].

The content of free sulfhydryl groups (-SH) and disulfide bonds (-S-S-) were important active groups in protein structure, and their changes reflected the protein oxidation and aggregation process [25,39]. The free sulfhydryl content of winged bean protein decreased significantly with increasing AAPH concentration: from 3.23 ± 0.03 μmol/g (AAPH = 0.00 mmol/L) to 1.39 ± 0.09 μmol/g (AAPH = 1.00 mmol/L). The total disulfide/sulfhydryl content also decreased with increasing AAPH concentration: from 68.47 ± 1.97 μmol/g (AAPH = 0.00 mmol/L) to 44.86 ± 1.89 μmol/g (AAPH = 1.00 mmol/L), indicating that AAPH-induced oxidation led to the consumption of free sulfhydryl groups within proteins, either through the formation of disulfide bonds (possibly more complex oxidative cross-linking or cleavage), or conversion to other oxidative products (such as sulfonic acid). Protein oxidation typically led to the conversion of sulfhydryl groups to disulfide bonds or further oxidation, thereby altering the protein’s folding state [25,40]. For example, oxidation of shrimp myofibrillar protein resulted in decreased sulfhydryl content. Additionally, incubation of soybean meal protein with AAPH also led to reduced sulfhydryl content [15].

### 3.4. Impacts of Peroxyl Radical-Induced Oxidative Modification on the Secondary Structure of Winged Bean Protein

As shown in Table 2, the α-helix content slightly increased to (2.40 ± 0.15) % at 0.04 mmol/L AAPH, reached its peak at (4.30 ± 0.21) % at 0.2 mmol/L AAPH, and then decreased to (2.05 ± 0.07) % at 1 mmol/L AAPH. The β-sheet content showed a trend of first decreasing and then increasing with the concentration of AAPH, dropping from (61.30 ± 0.32) % (AAPH = 0.00 mmol/L) to (45.75 ± 0.31) % (AAPH = 0.20 mmol/L), and subsequently increasing to 56.40 ± 0.09% when the AAPH concentration reached 1 mmol/L. The random coil content initially increased and then decreased, rising from (37.30 ± 0.21) % in the control group (AAPH = 0.00 mmol/L) to (49.90 ± 0.18) % at 0.2 mmol/L AAPH, and finally dropping to (41.55 ± 0.07) % at 1 mmol/L AAPH.

The content of β-turn structure in winged bean protein was relatively low and remained almost unchanged. These secondary structural changes indicate that AAPH-induced oxidation led to significant rearrangement of protein conformation. β-sheets were the major secondary structural component of winged bean protein, which was consistent with the characteristics of many legume proteins [41,42]. The decrease in β-sheet content and the increase in random coil content typically indicated protein unfolding or structural loosening [15,43,44]. This unfolding might expose more hydrophobic groups (explaining the increased surface hydrophobicity of winged bean protein when 0.04 mmol/L AAPH was added in this experiment). However, at higher AAPH concentrations, although the random coil content decreased and the β-sheet content recovered, it remained lower than that of the blank group (AAPH = 0.00 mmol/L), which might suggest the formation of new, more compact (but non-native) aggregated structures, or enhanced hydrophobic interactions within the protein, but not a complete restoration to the original folded state.

### 3.5. Impacts of Peroxyl Radical-Induced Oxidative Modification on the Tertiary Structure of Winged Bean Protein

#### 3.5.1. Intrinsic Fluorescence Spectrum

The maximum emission wavelength (λ_max_) and fluorescence intensity (peak intensity) of intrinsic fluorescence spectra can sensitively reflect changes in the microenvironment of protein-internal tryptophan residues, serving as an effective method for detecting changes in protein tertiary structure [28,45].

As shown in Figure 6 and Table 3, when the AAPH concentration was 0.00 mmol/L, the wing bean protein exhibited a fluorescence peak (λ_max_) at 346.2 nm with an intensity of 2487 a.u. When the AAPH concentration increased to 0.04 mmol/L, the λ_max_ of wing bean protein shifted to 339.4 nm (blue shift), and the peak intensity significantly increased to 3520 a.u., indicating moderate protein unfolding and exposure of tryptophan residues to a more hydrophobic environment. Notably, this finding complemented the experimental result of a substantial increase in surface hydrophobicity (from 847.48 to 2451.70). These two experimental indicators collectively revealed: (1) The protein might have undergone moderate conformational adjustments, which, although placing tryptophan residues in relatively hydrophobic environments, reduced the presence of dynamic quenchers (such as water molecules and oxygen) around them, leading to enhanced fluorescence. (2) Meanwhile, the increased surface hydrophobicity reflected the exposure of hydrophobic regions containing non-tryptophan residues [46].

When the AAPH concentration was 0.2 mmol/L, the λ_max_ shifted to 348.6 nm and the peak intensity decreased to 2246 a.u. This was due to protein aggregation, where tryptophan residues might be reburied in a more polar environment or undergo oxidative modification, leading to fluorescence quenching. This corresponded with the decrease in surface hydrophobicity (from 2451.70 to 1367.60), increased carbonyl content, and larger average particle size in this study. When the AAPH concentration reached 1.00 mmol/L, the λ_max_ shifted back to 342.8 nm, and the peak intensity slightly increased to 2418 a.u. At this AAPH concentration, the solubility of winged bean protein showed a slight improvement, suggesting some degree of rearrangement or fragmentation of the protein under extreme oxidative conditions, with the formation of soluble structures from large molecular aggregates.

#### 3.5.2. Raman Spectrum

##### Analysis of Typical Functional Groups and Carbon-Hydrogen Bonds

Raman spectroscopy was an effective tool for characterizing the secondary and tertiary structures of proteins, particularly sensitive to the vibrational modes of aromatic amino acids such as tryptophan, tyrosine, phenylalanine, and C-H bonds.

The Fermi resonance double peaks near 830 cm^−1^ and 850 cm^−1^ were contributed by tyrosine residues. The intensity ratio (I_850/830_) of these peaks could indicate the state of tyrosine in protein molecules [47]. As shown in Figure 7 and Table 4, when the AAPH concentration was 0.00 mmol/L, I_850/830_ was 1.13, indicating that under non-oxidized conditions, tyrosine residues in winged bean seed protein were exposed in a relatively polar environment or simultaneously act as both weak hydrogen bond donors and acceptors. As the AAPH concentration increased to 0.04 mmol/L and 0.20 mmol/L, the I_850/830_ values decreased to 0.94 and 0.90, respectively. The I_850/830_ between 0.70 and 1.00 typically suggests that tyrosine residues were embedded in hydrophobic environments or acted as hydrogen bond donors. This was likely due to protein conformational changes under low-concentration AAPH oxidation, causing some tyrosine residues to shift from hydrophilic surfaces to hydrophobic regions inside the protein, or changes in their hydrogen bonding status, forming stronger hydrogen bonds [48].

When the AAPH concentration was further increased to 1.00 mmol/L, the I_850/830_ increased to 1.39. This significant increase was likely due to more severe protein structure damage under high-intensity oxidative conditions, leading to the re-exposure of tyrosine residues originally buried in the hydrophobic core to a polar solvent environment, or the disruption of their hydrogen bonding networks, which enhanced their characteristics as weak hydrogen bond donors and acceptors. Protein oxidation could lead to protein denaturation and aggregation, thereby altering the microenvironment of aromatic amino acids [49,50]. In summary, under low oxidation conditions (AAPH ≤ 0.20 mmol/L), tyrosine formed stronger hydrogen bonds (e.g., with adjacent carbonyl or amino groups), which enhanced its properties in hydrophobic environments. However, under high oxidation conditions (AAPH = 1.00 mmol/L), these hydrogen bond stabilities were disrupted, and the damage to hydrogen bonds accompanied the aggregation of hydrophobic residues, thus promoting the formation of β-sheet structures and protein aggregation behavior. The circular dichroism and laser particle size analysis data in this experiment strongly supported this conclusion.

The changes in Raman peak intensity of tryptophan indole ring were closely related to protein structural changes, especially the exposure degree of tryptophan residues [51]. When the Raman peak intensity of tryptophan indole ring increased, it indicated that tryptophan residues had changed from a buried state to an exposed state; conversely, a decrease in peak intensity might suggest reduced intermolecular binding of proteins, protein unfolding, or subunit separation, with tryptophan residues being more buried within the molecular interior.

When the AAPH concentration was 0.00 mmol/L, 0.04 mmol/L, 0.2 mmol/L, and 1.00 mmol/L, the tryptophan indole ring peak intensity was 0.91, 0.60, 0.66, and 0.62, respectively. The decreasing trend from 0.91 to 0.60 suggested that at lower AAPH concentrations (AAPH = 0.04 mmol/L), tryptophan residues might have undergone a process from relative exposure to burial, which was related to the conformational adjustment or partial folding of proteins in response to oxidative stress. Subsequently, when the AAPH concentration was 0.2 mmol/L and 1.00 mmol/L, the Raman peak intensity remained at a lower level (0.66 and 0.62), possibly due to the continuous burial of tryptophan residues or deeper rearrangement of protein conformation to protect these susceptible residues from oxidation. This decreasing trend also implied the protein unfolding process (the secondary structure composition and content analysis in this experiment also showed a decrease in regular secondary structure units and a significant increase in random coil structure), with tryptophan residues gradually becoming further buried. This conclusion was consistent with the results of the previous intrinsic fluorescence spectroscopy.

The peak intensity changes of aliphatic C-H bending vibrations could reflect the flexibility or compactness of the overall protein backbone [52,53]. In the unoxidized state, the C-H bending vibration peak intensity was 1.11. As the AAPH concentration increased, the C-H bending vibration peak intensity decreased, reaching 0.93 and 0.91 at 0.04 mmol/L and 0.20 mmol/L, respectively. This suggested that the flexibility of the protein backbone increased, or the structure became looser, leading to changes in the vibrational environment of C-H groups. When the AAPH concentration was 1.00 mmol/L, the peak intensity slightly increased to 1.01. This might indicate that under extreme oxidation conditions, protein molecules might aggregate, thereby restricting the movement of aliphatic C-H groups again, or forming new compact structures. This conclusion was consistent with the detection results of secondary structure composition and content in this experiment.

The decrease in peak intensity of aliphatic C-H stretching vibrations was typically associated with the disruption or exposure of protein hydrophobic cores [52]. When the AAPH concentration was 0.00 mmol/L, the peak intensity of aliphatic amino acid C-H stretching vibrations was 2.19. As the AAPH concentration increased to 0.04 mmol/L and 0.2 mmol/L, the peak intensity decreased to 1.83 and 1.61, respectively, indicating disruption or exposure of protein hydrophobic cores. Oxidative stress could lead to protein denaturation and exposure of hydrophobic regions, thereby promoting aggregation. When the AAPH concentration reached 1.00 mmol/L, the peak intensity slightly increased to 1.93. This was similar to the increasing trend of the I_850/830_, which might be due to high-intensity oxidation causing proteins to undergo denaturation, aggregation, and even partial hydrolysis, resulting in hydrophobic groups rearranging in new conformations or through intermolecular aggregation, thus affecting their vibrational intensity. Previous studies had reported that protein oxidation could lead to protein denaturation and the formation of modified products such as carbonyls [3].

##### Disulfide Bond Configuration

Disulfide bonds played a crucial role in stabilizing the tertiary structure of proteins. Their conformations were commonly classified as: gauche-gauche-gauche (g-g-g, strongly hydrophobic, usually found in the protein interior), gauche-gauche-trans (g-g-t, moderately hydrophobic), and trans-gauche-trans (t-g-t, hydrophilic, typically located on the surface) [54].

As shown in Figure 8 and Table 5, with increasing AAPH concentration, the proportion of g-g-g conformation decreased significantly from 55.1% to 30.8%, while the proportions of g-g-t and t-g-t conformations increased, particularly the t-g-t conformation, which increased from 9.5% to 19.9%. This indicated that oxidation led to the loosening of tightly packed disulfide bond structures within proteins, with some disulfide bonds shifting from hydrophobic interiors to more hydrophilic surfaces or undergoing conformational changes [55,56,57]. This was consistent with the decrease in β-sheet structure and increase in random coil observed in winged bean protein in this study, reflecting increased protein unfolding and structural flexibility. The decrease in total disulfide/thiol content also supported the oxidative disruption or rearrangement of disulfide bonds. Oxidative stress could lead to disulfide bond cleavage or the formation of mixed disulfides, thereby affecting the overall protein stability.

## 4. Conclusions

When the concentration of AAPH (0.04 mmol/L) was low, the surface hydrophobicity of winged bean protein significantly increased due to the exposure of hydrophobic regions of non-tryptophan residues caused by oxidative modification, while the absolute value of zeta potential decreased, solubility moderately improved, protein degradation occurred, and particle size decreased. When AAPH concentration reached 0.20 mmol/L, winged bean protein aggregates formed due to oxidation, resulting in decreased surface hydrophobicity and solubility. At an AAPH concentration of 1.00 mmol/L, further oxidation of winged bean protein occurred, leading to a further decrease in surface hydrophobicity. Protein aggregates rearranged, forming more soluble aggregates, and solubility increased again.

Overall, the addition of AAPH had little effect on the β-turn structure and solubility of winged bean protein. When the concentration of AAPH increased from 0.00 mmol/L to 1.00 mmol/L, both the α-helix structure and random coil structure showed a trend of first increasing and then decreasing, while the β-sheet structure showed a trend of first decreasing and then increasing. The Raman spectroscopy characterization results were in good agreement with other data in this study. This study had important implications for the application of winged bean protein in food processing and formulation: under mild oxidation conditions (AAPH = 0.04 mmol/L), the functional properties of winged bean protein could be optimized. However, under relatively strong oxidation conditions (AAPH ≥ 0.20 mmol/L), the structural integrity and functionality of winged bean protein would be compromised, requiring special attention during food storage and processing.

In general, from an industrial perspective, these findings established a scientific foundation for the rational utilization of winged bean protein in food manufacturing. Mild oxidation (0.04 mmol/L AAPH) could be strategically utilized to tailor the functional properties of winged bean protein for specific food formulations, while strict control of oxidative stress (avoiding exposure to ≥0.20 mmol/L AAPH-equivalent conditions) was essential to maintain the structural and functional stability of the protein during industrial storage and processing.

## Figures and Tables

**Figure 1 foods-14-04120-f001:**
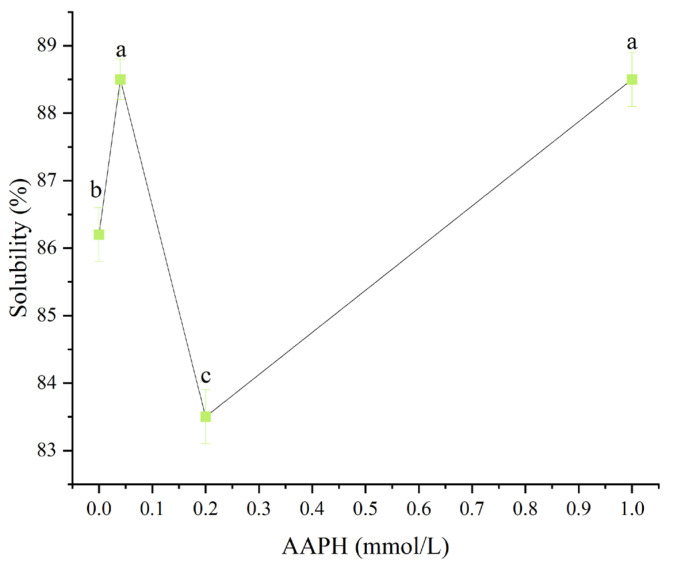
Effects of different concentrations of AAPH (0.00 mmol/L, 0.04 mmol/L, 0.20 mmol/L, 1.00 mmol/L) on the solubility of winged bean protein. Each datapoint is the mean and standard deviation of five measurements. Different letters (a–c) indicate significant differences (*p* < 0.05).

**Figure 2 foods-14-04120-f002:**
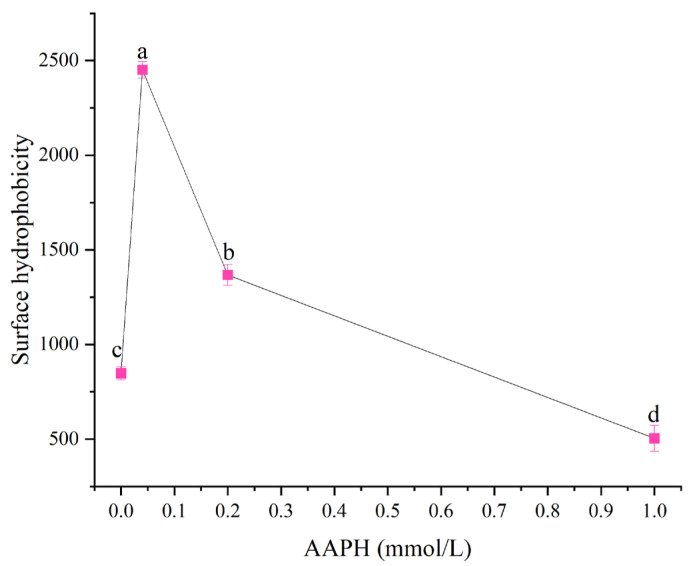
Effects of different concentrations of AAPH (0.00 mmol/L, 0.04 mmol/L, 0.20 mmol/L, 1.00 mmol/L) on the surface hydrophobicity of winged bean protein. Each datapoint is the mean and standard deviation of five measurements. Different letters (a–d) indicate significant differences (*p* < 0.05).

**Figure 3 foods-14-04120-f003:**
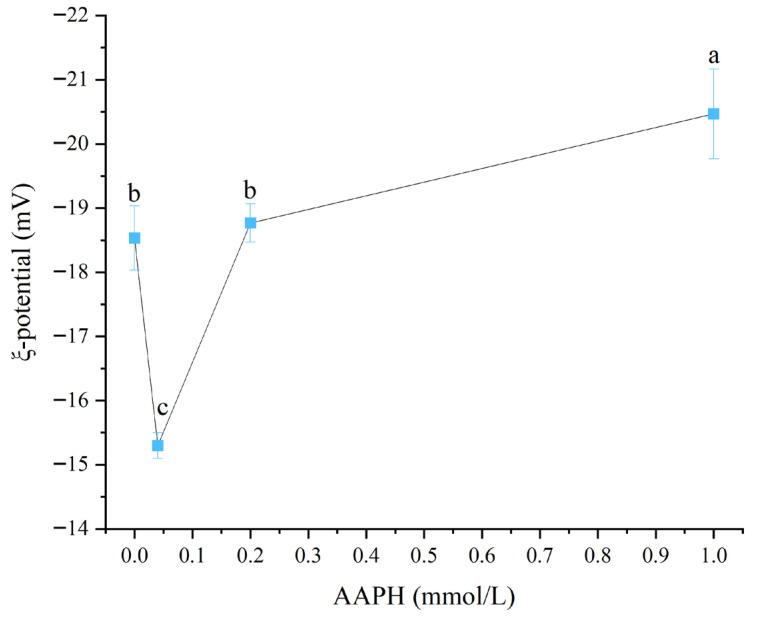
Effects of different concentrations of AAPH (0.00 mmol/L, 0.04 mmol/L, 0.20 mmol/L, 1.00 mmol/L) on the zeta potential of winged bean protein. Each datapoint is the mean and standard deviation of five measurements. Different letters (a–c) indicate significant differences (*p* < 0.05).

**Figure 4 foods-14-04120-f004:**
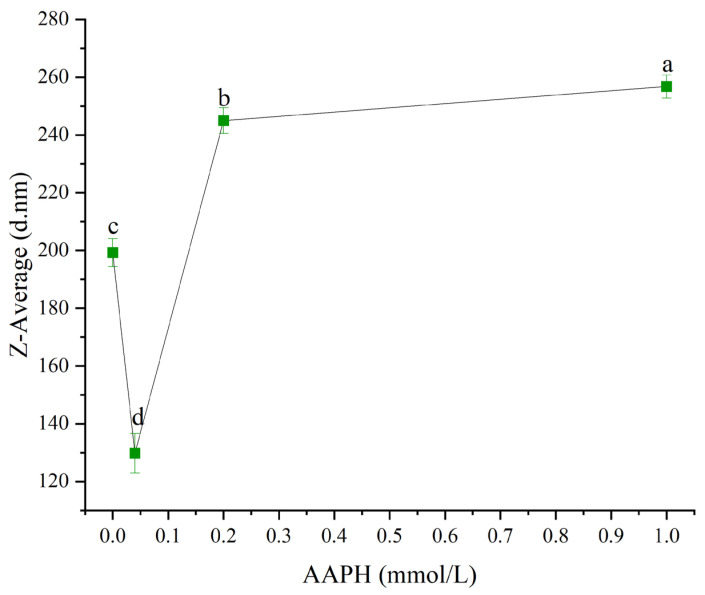
Effects of different concentrations of AAPH (0.00 mmol/L, 0.04 mmol/L, 0.20 mmol/L, 1.00 mmol/L) on the mean particle size of winged bean protein. Each datapoint is the mean and standard deviation of five measurements. Different letters (a–d) indicate significant differences (*p* < 0.05).

**Figure 5 foods-14-04120-f005:**
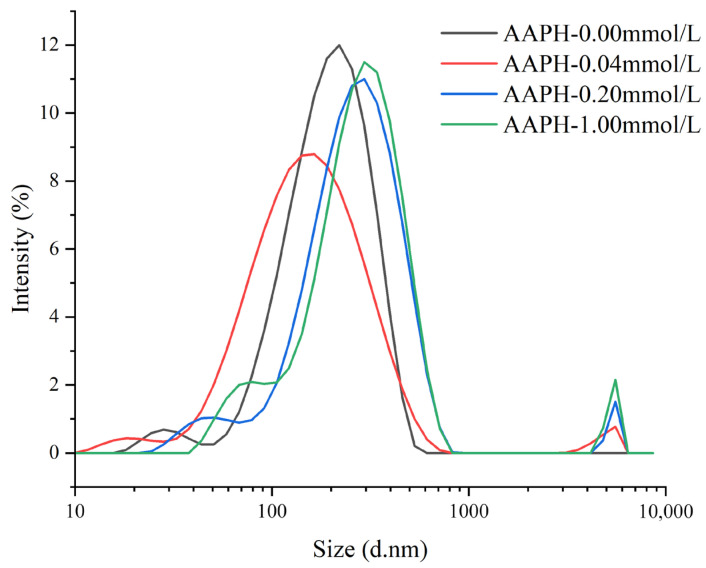
Effects of different concentrations of AAPH (0.00 mmol/L, 0.04 mmol/L, 0.20 mmol/L, 1.00 mmol/L) on the particle size distribution of winged bean protein.

**Figure 6 foods-14-04120-f006:**
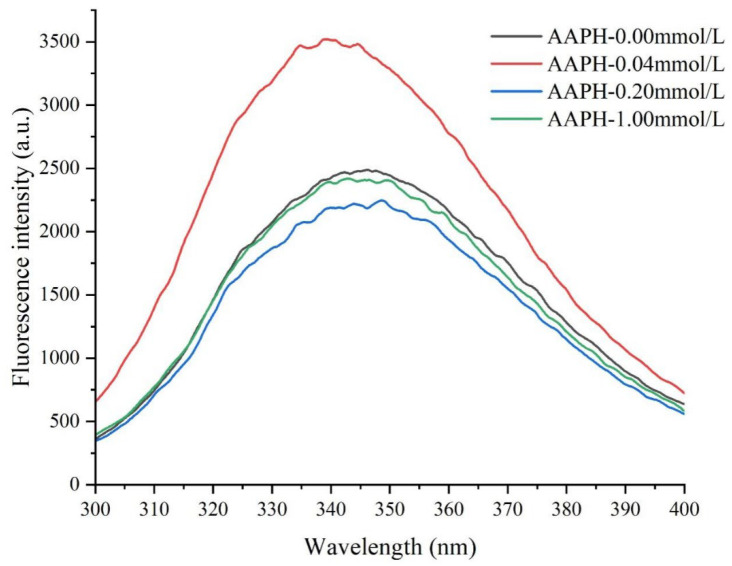
Effects of different concentrations of AAPH (0.00 mmol/L, 0.04 mmol/L, 0.20 mmol/L, 1.00 mmol/L) on the intrinsic fluorescence of winged bean protein.

**Figure 7 foods-14-04120-f007:**
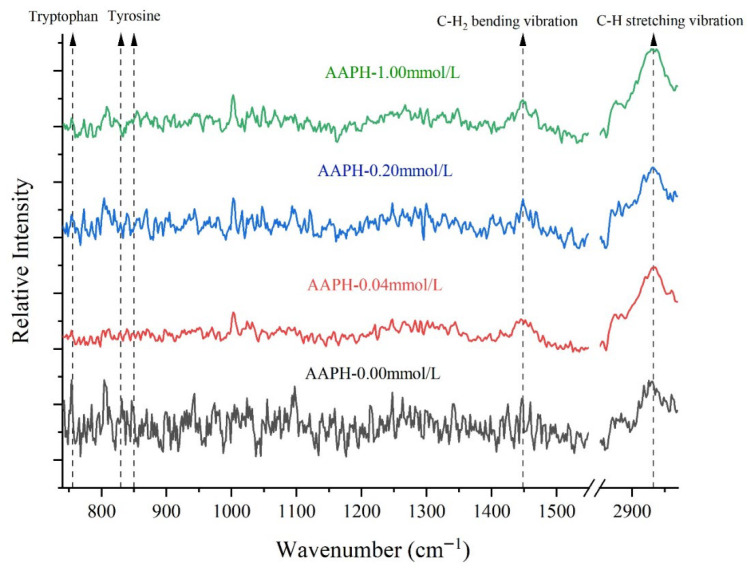
Effects of different concentrations of AAPH (0.00 mmol/L, 0.04 mmol/L, 0.20 mmol/L, 1.00 mmol/L) on FT-Raman spectra of winged bean protein.

**Figure 8 foods-14-04120-f008:**
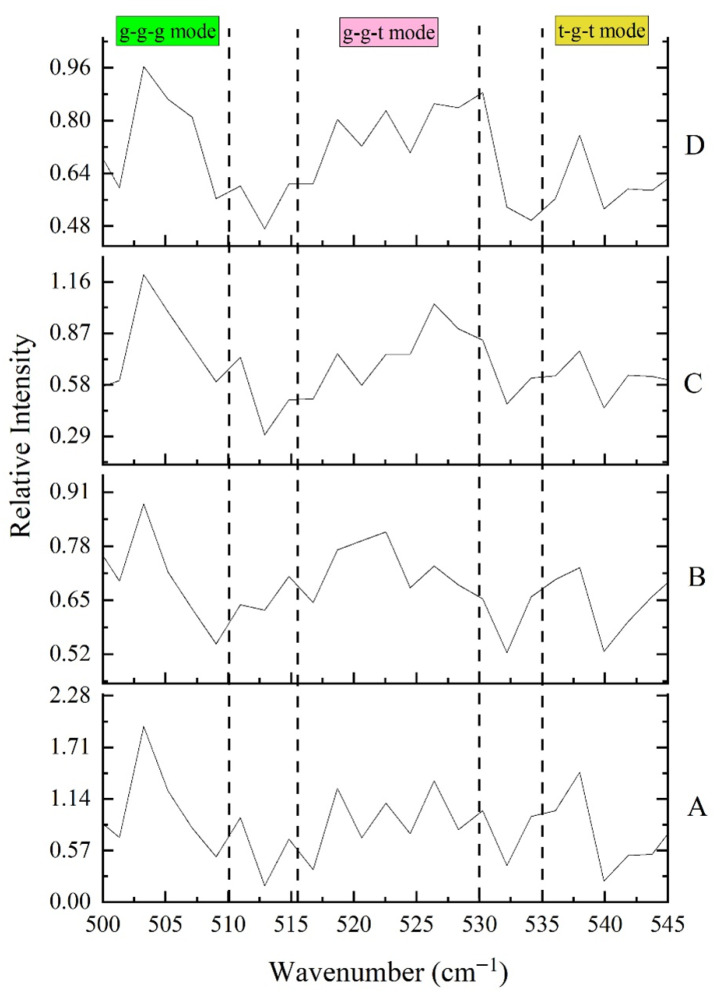
Effects of different concentrations of AAPH (0.00 mmol/L, 0.04 mmol/L, 0.20 mmol/L, 1.00 mmol/L) on the disulfide bond configuration of winged bean protein. A: AAPH = 0.00 mmol/L; B: AAPH = 0.04 mmol/L; C: AAPH = 0.20 mmol/L; D: AAPH = 1.00 mmol/L.

**Table 1 foods-14-04120-t001:** Impacts of gradient AAPH concentrations (0.00, 0.04, 0.20, and 1.00 mmol/L) on carbonyl content, free sulphydryl groups, and disulphide bond levels in winged bean protein.

AAPH (mmol/L)	Protein Carbonyl (μmol/g)	Free Sulphydryl (μmol/g)	Total Disulphide and Sulphydryl (μmol/g)
0.00	1.75 ± 0.11 D	3.23 ± 0.03 A	68.47 ± 1.97 A
0.04	2.17 ± 0.12 C	2.92 ± 0.05 B	60.02 ± 1.66 B
0.20	3.06 ± 0.09 B	2.19 ± 0.05 C	52.33 ± 2.09 C
1.00	4.22 ± 0.13 A	1.39 ± 0.09 D	44.86 ± 1.89 D

Each data is the means and standard deviations of five measurements. Different letters (A–D) in the same column indicate significant differences (*p* < 0.05).

**Table 2 foods-14-04120-t002:** Effects of different concentrations of AAPH (0.00 mmol/L, 0.04 mmol/L, 0.20 mmol/L, 1.00 mmol/L) on secondary structure composition and content of winged bean protein.

AAPH (mmol/L)	Structural Composition and Content (%)			
	α-Helix	β-Sheet	β-Turn	Random Coil
0.00	1.40 ± 0. 14 D	61.30 ± 0.32 A	0.00 ± 0.00 B	37.30 ± 0.21 C
0.04	2.40 ± 0.15 B	59.90 ± 0.37 B	0.15 ± 0.05 A	37.50 ± 0.25 C
0.20	4.30 ± 0.21 A	45.75 ± 0.31 D	0.07 ± 0.04 A	49.90 ± 0.18 A
1.00	2.05 ± 0.07 C	56.40 ± 0.09 C	0.00 ± 0.00 B	41.55 ± 0.07 B

Each datapoint is the mean and standard deviation of five measurements. Different letters (A–D) in the same column indicate significant differences (*p* < 0.05).

**Table 3 foods-14-04120-t003:** Effects of different concentrations of AAPH (0.00 mmol/L, 0.04 mmol/L, 0.20 mmol/L, 1.00 mmol/L) on the intrinsic fluorescence spectrum of winged bean protein.

AAPH (mmol/L)	Intrinsic Fluorescence Spectrum	
	λ_max_ (nm)	FI (a.u.)
0.00	346.2 ± 0.4 B	2487.5 ± 15.9 B
0.04	339.4 ± 0.5 D	3520.3 ± 12.5 A
0.20	348.6 ± 0.8 A	2246.6 ± 18.4 D
1.00	342.8 ± 0.5 C	2418.8 ± 17.2 C

Each datapoint is the mean and standard deviation of five measurements. Different letters (A–D) in the same column indicate significant differences (*p* < 0.05).

**Table 4 foods-14-04120-t004:** Normalized intensity values at selected regions of the FT-Raman spectra of winged bean protein treated by different concentrations of AAPH (0.00 mmol/L, 0.04 mmol/L, 0.20 mmol/L, 1.00 mmol/L).

Band Assignment [Wavenumber (cm^−1^)]	Normalized Intensity Values			
	AAPH-0.00 mmol/L	AAPH-0.04 mmol/L	AAPH-0.20 mmol/L	AAPH-1.00 mmol/L
Tyrosine residues vibration [I_850/830_]	1.13 ± 0.08 b	0.94 ± 0.05 c	0.90 ± 0.04 c	1.39 ± 0.06 a
Tryptophan indole ring [756~758 cm^−1^]	0.91 ± 0.02 a	0.60 ± 0.03 c	0.68 ± 0.02 b	0.62 ± 0.02 c
Aliphatic residues C-H_2_ bending vibration [1446~1448 cm^−1^]	1.11 ± 0.03 a	0.93 ± 0.02 c	0.91 ± 0.02 c	1.01 ± 0.04 b
Aliphatic residues C-H stretching vibration [2931~2933 cm^−1^]	2.19 ± 0.03 a	1.83 ± 0.04 c	1.61 ± 0.02 d	1.93 ± 0.03 b

Each datapoint is the mean and standard deviation of five measurements. Different letters (a–d) in the same row indicate significant differences (*p* < 0.05).

**Table 5 foods-14-04120-t005:** Disulfide bond configuration of winged bean protein treated by different concentrations of AAPH (0.00 mmol/L, 0.04 mmol/L, 0.20 mmol/L, 1.00 mmol/L).

AAPH	Disulfide Bond Configuration		
	g-g-g (%)	g-g-t (%)	t-g-t (%)
0.00	55.1 ± 0.2 A	35.4 ± 0.3 D	9.5 ± 0.2 D
0.04	33.3 ± 0.3 C	55.2 ± 0.2 A	11.5 ± 0.2 C
0.20	39.1 ± 0.2 B	46.4 ± 0.3 C	14.5 ± 0.3 B
1.00	30.8 ± 0.2 D	49.3 ± 0.2 B	19.9 ± 0.3 A

Each datapoint is the mean and standard deviation of five measurements. Different letters (A–D) in the same column indicate significant differences (*p* < 0.05).

## Data Availability

The original contributions presented in this study are included in the article. Further inquiries can be directed to the corresponding author.

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
