# Peer review of "Effects of 2,2′-Azobis(2-Amidinopropane) Dihydrochloride (AAPH) on Functional Properties and Structure of Winged Bean Protein"

_foods, 2025, doi:10.3390/foods14234120_

Round 1

Reviewer 1 Report

Comments and Suggestions for Authors

This manuscript, titled “Effects of 2,2′-Azobis(2-Amidinopropane) Dihydrochloride (AAPH) on Functional Properties and Structure of Winged Bean Protein presents an interesting research paper. This study uses a variety of analytical techniques (solubility, zeta potential, DLS, FT-Raman, intrinsic fluorescence, and CD spectroscopy) to examine the structural and functional impacts of AAPH-induced oxidative alteration on winged bean protein isolate (WPI). Given the growing interest in innovative plant-based proteins and their oxidative stability in food systems, the topic is relevant today.

-Line 17-19; summarize methods that used for investigation of functional properties.

-Include more details regarding to Winged Bean, and importance.

-To enable replication, specify the protein concentration used in each structural analysis (CD, Raman, fluorescence).

-Line 62-65; explain biochemical mechanism behind the idea?

-Line 67-70; discuss the reports that mentioned regarding to effects of AAPH on winged bean protein.

-Line 369, please consider style of references.

-In order ensure correlation validity, make clear if the same samples were examined in each experiment.

-Researchers indicated that “Notably, when the AAPH concentration was 0.2 mmol/L, the solubility of winged bean protein decreased slightly to 83.5% (p<0.05). This decrease in solubility might be related to protein aggregation, conformational changes, surface hydrophobicity, and Zeta potential”. The explanation for the decreased solubility at 0.2 mmol/L AAPH (lines 207–210) is overly general. The researchers should provide a more detailed mechanistic informations.

-Expand the discussion of Raman spectral changes (I850/830 ratio) with structural implications more specifically, how oxidation modifies tyrosine hydrogen-bond networks and contributes to β-sheet rearrangement.

-Include a brief overview on any possible industrial outcomes for the conclusion section

Author Response

1)Reviewer #1: Line 17-19; summarize methods that used for investigation of functional properties.

Our reply:

We are much obliged to the reviewer for his constructive and helpful comments. We have added specific testing metrics into the paper using a red typeface according to the reviewer's comment. (Line 17-18)

2)Reviewer #1: Include more details regarding to Winged Bean, and importance.

Our reply:

We have made the revisions in a red typeface according to the reviewer's comments. (Line 41-43)

3)Reviewer #1: To enable replication, specify the protein concentration used in each structural analysis (CD, Raman, fluorescence).

Our reply:

We have comprehensively adopted the reviewer's comments. First, regarding the CD and fluorescence tests, which were conducted using solution samples, the concentration of the test samples was already stated in my original manuscript, which the reviewer might have overlooked. Therefore, no additional concentration information was added in the revised manuscript. Second, for the Raman spectroscopy testing, we used solid powder samples. We have accepted the reviewer's suggestion and explicitly stated in the text that the samples are solid powder samples using a red typeface. (Line 210)

4)Reviewer #1: Line 62-65; explain biochemical mechanism behind the idea?

Our reply:

We have made the revisions in a red typeface according to the reviewer's comments. (Line 65-67)

5)Reviewer #1: Line 67-70; discuss the reports that mentioned regarding to effects of AAPH on winged bean protein.

Our reply:

I have searched the SCI database and have not yet found any publicly published studies on the effects of AAPH on winged bean protein. This further demonstrates the novelty and necessity of this research paper.

6)Reviewer #1: Line 369, please consider style of references

Our reply:

We have made the revisions in a red typeface according to the reviewer's comments. (Line 399)

7)Reviewer #1: In order ensure correlation validity, make clear if the same samples were examined in each experiment.

Our reply:

The winged bean protein isolate used in this study was prepared according to the method described in section "2.2. Preparation of Lipid Free-Winged Bean Protein Isolate" of this paper. The preparation process strictly followed the operational procedures to ensure the scientific nature of sample preparation and the stability of protein characteristics.

8)Reviewer#1: Researchers indicated that “Notably, when the AAPH concentration was 0.2 mmol/L, the solubility of winged bean protein decreased slightly to 83.5% (p<0.05). This decrease in solubility might be related to protein aggregation, conformational changes, surface hydrophobicity, and Zeta potential”. The explanation for the decreased solubility at 0.2 mmol/L AAPH (lines 207–210) is overly general. The researchers should provide a more detailed mechanistic informations.

Our reply:

We are much obliged to the reviewer for his constructive and helpful comments. We have made the revisions in a red typeface according to the reviewer's comments. (Line 244-247)

9)Reviewer#1: Expand the discussion of Raman spectral changes (I850/830 ratio) with structural implications more specifically, how oxidation modifies tyrosine hydrogen-bond networks and contributes to β-sheet rearrangement.

Our reply:

We are much obliged to the reviewer for his constructive and helpful comments. We have made the revisions in a red typeface according to the reviewer's comments. (Line 511-518)

10)Reviewer#1: Include a brief overview on any possible industrial outcomes for the conclusion section

Our reply:

We are much obliged to the reviewer for his constructive and helpful comments. We have made the revisions in a red typeface according to the reviewer's comments. (Line 614-620)

Reviewer 2 Report

Comments and Suggestions for Authors

The manuscript “Effects of 2,2’-Azobis Dihydrochloride on Functional Properties and Structure of Winged Bean Protein” by Fang et al. aimed to understand how free radical oxidation (using AAPH) affects the structure and function of winged bean protein. In this study, the authors observed that the effects of AAPH on the Winged Bean Protein are concentration-dependent: mild oxidation can improve winged bean protein's functionality, but stronger oxidation damages it. This study helps clarify the oxidative modification mechanism of this protein for food processing applications. 

The study is interesting, and the authors provide detailed methods about the study. But there are some concerns the authors should address shown below:

1. In figure 1: it is highly recommended that the authors directly label x-axis as 0.00, 0.04, 0.20, 1.00, the name of x-axis should be “Concentration of AAPH (mmol/L)”. Similarly, please update figure 2-4 as well. 

2. In figure 5: in panel 2 (blue color): the y-axis should start from “0.0” instead of “10.0”; It is highly suggested that the authors change the figure legend: AAPH = 0.00 mmol/L, AAPH = 0.04 mmol/L, AAPH = 0.20 mmol/L, AAPH = 1.00 mmol/L

3. Since the y-axis is consistent, the authors should merge four different curves into one figure, to show the difference of Size (d.nm.) among them. 

4. In table 1, the authors should use upper letter (a, b, c,)  to label the statistical significance in the columns instead of “1.75±0.11d”. Similarly, in table 2, table 3, table 4, and table 5

5. In line 370: “... with increasing…” should be “... with decreasing …”, and add “ (AAPH=1.00 mmol/L)” at the end of text. 

Author Response

1)Reviewer #2: In figure 1: it is highly recommended that the authors directly label x-axis as 0.00, 0.04, 0.20, 1.00, the name of x-axis should be “Concentration of AAPH (mmol/L)”. Similarly, please update figure 2-4 as well.

Our reply:

We are much obliged to the reviewer for his constructive and helpful comments. We have redrawn Figures 1, 2, 3, and 4 according to the reviewer's suggestions, and revised the corresponding figure captions (Line 249-251, Line 264-266, Line 301-303, Line 327-329)

2)Reviewer #2: (1) In figure 5: in panel 2 (blue color): the y-axis should start from “0.0” instead of “10.0”; It is highly suggested that the authors change the figure legend: AAPH = 0.00 mmol/L, AAPH = 0.04 mmol/L, AAPH = 0.20 mmol/L, AAPH = 1.00 mmol/L; (2) Since the y-axis is consistent, the authors should merge four different curves into one figure, to show the difference of Size (d.nm.) among them.

Our reply:

We are much obliged to the reviewer for his constructive and helpful comments. We have redrawn Figures 5 according to the reviewer's suggestions.

3)Reviewer #2: In table 1, the authors should use upper letter (a, b, c,)  to label the statistical significance in the columns instead of “1.75±0.11d”. Similarly, in table 2, table 3, table 4, and table 5

Our reply:

We are much obliged to the reviewer for his constructive and helpful comments. We have used upper letter (A, B, C, D) to label the statistical significance in the columns according to the reviewer's suggestions, and revised the corresponding table captions (Line 395-396, Line 425-426, Line 465-466, Line 589-590).

Table 4 remains unchanged because different letters (a-d) in the same row indicate significant differences (p <0.05).

4)Reviewer #2: In line 370: “... with increasing…” should be “... with decreasing …”, and add “ (AAPH=1.00 mmol/L)” at the end of text.

Our reply:

We are much obliged to the reviewer for his constructive and helpful comments. We have made the revisions in a red typeface according to the reviewer's comments. (Line 401)